

# An enhanced lightweight T-Net architecture based on convolutional neural network (CNN) for tomato plant leaf disease classification

Amreen Batool[1], Jisoo Kim[2], Sang-Joon Lee[3], Ji-Hyeok Yang[4] and Yung-Cheol Byun[5]

[1] Electronic Engineering, Jeju National University, Jeju, Republic of South Korea
[2] Institute of Information Science & Technology, Jeju National University, Jeju, Republic of South Korea
[3] Department of Computer Engineering, Jeju National University, Jeju, Republic of South Korea
[4] Nanoom Energy Co. Ltd, Jeju-si, Jeju-do, Republic of South Korea
[5] Computer Engineering/ Electronic Engineering, Jeju National University, Jeju, Republic of South Korea

Corresponding author
Yung-Cheol Byun, ycb@jejunu.ac.kr

## ABSTRACT

Tomatoes are a widely cultivated crop globally, and according to the Food and Agriculture Organization (FAO) statistics, tomatoes are the third after potatoes and sweet potatoes. Tomatoes are commonly used in kitchens worldwide. Despite their popularity, tomato crops face challenges from several diseases, which reduce their quality and quantity. Therefore, there is a significant problem with global agricultural productivity due to the development of diseases related to tomatoes. Fusarium wilt and bacterial blight are substantial challenges for tomato farming, affecting global economies and food security. Technological breakthroughs are necessary because existing disease detection methods are time-consuming and labor-intensive. We have proposed the T-Net model to find a rapid, accurate approach to tackle the challenge of automated detection of tomato disease. This novel deep learning model utilizes a unique combination of the layered architecture of convolutional neural networks (CNNs) and a transfer learning model based on VGG-16, Inception V3, and AlexNet to classify tomato leaf disease. Our suggested T-Net model outperforms earlier methods with an astounding 98.97% accuracy rate. We prove the effectiveness of our technique by extensive experimentation and comparison with current approaches. This study offers a dependable and understandable method for diagnosing tomato illnesses, marking a substantial development in agricultural technology. The proposed T-Net-based framework helps protect crops by providing farmers with practical knowledge for managing disease. The source code can be accessed from the given link.

## INTRODUCTION

Tomatoes are widely grown and used around the world, and they are also full of essential nutrients. In early 2024, the Food and Agriculture Organization (FAO) reported that

millions of tons of tomatoes were produced globally (*FAO, 2024*). Tomato growing is a mainstay of agriculture; various diseases constantly threaten this vital crop, endangering harvests and food security. In the tomato growing industry, state-of-the-art technology, such as CNN models, has changed the game regarding disease identification and control in recent years. There are a variety of minerals and phytochemicals in tomatoes, including lycopene, potassium, iron, folate, and vitamin C (*Borguini & Ferraz Da Silva Torres, 2009*). Tomatoes are a popular vegetable widely farmed throughout the world and a good source of income for growers (*Mba et al., 2024*). In addition to their nutritional value, tomatoes are an excellent addition to a balanced diet because they can be consumed raw or cooked without losing any dietary qualities (*Kang, 2023*). Over 80% of commercially grown tomatoes produce processed goods such as ketchup, soup, and juice (*Viuda-Martos et al., 2014*). There are several health benefits associated with tomatoes, many attributed to their high antioxidant content (*Rao & Agarwal, 1999*). The article discusses the effects of growing conditions on tomato cultivars and their potential health benefits.

Plant diseases that cause considerable harm to agricultural yield worldwide include Fusarium wilt, bacterial blight, and powdery mildew. Early disease identification and categorization of tomato plants can reduce the need for costly crop treatments and thus increase food production for farmers. The plant the pathogen has infected is the host disease from the simultaneous occurrence of these components (*Ahmad, Saraswat & El Gamal, 2023*). The classification of tomato plant diseases has been the subject of substantial research. However, the similarity between affected and healthy leaves makes it challenging to locate and identify diseases promptly. While bacterial blight results in black lesions and wilting, fusarium wilt produces yellowing and withering of the leaves (*Sreedevi & Manike, 2024*). Powdery mildew creates white, powdery spots. Adequate disease control is essential to safeguard food security and preserve agricultural productivity. Certain unique circumstances can cause plant diseases. The link between three crucial elements, the environment, the host, and the infectious agent, is described explicitly by a conceptual model called the disease triangle. The disease does not manifest if any of these three elements is missing, and the triangle remains incomplete (*Thangaraj et al., 2022*). Abiotic elements that can significantly impact the plant include watering, pH, humidity, temperature, and airflow (*Chamard et al., 2024*). An organism that assaults a plant, such as a virus, bacterium, or fungus, is known as an infectious agent. Diseases often cause symptoms that damage the plant from the bottom up, and many spread very fast after getting an infection from other affected plants (*Khanday et al., 2024*). In addition, sensing-based technologies are utilized to monitor banana leaf diseases and classify a maturity level to enhance production quality (*Wang et al., 2023a*; *Wang et al., 2023b*; *Zhang et al., 2024b*).

Furthermore, the mosaic virus, yellow leaf curl virus, target spot, two-spotted spider mite, septoria leaf spot, leaf mold, late blight, early blight, and bacterial spot are a few of the most prevalent diseases that damage tomato leaves, as seen in Fig. 1. Recently, many technologies, including image processing pattern recognition and computer vision, have advanced fast and secure use in agriculture, particularly in the automation of pest and disease detection procedures (*Devaraj et al., 2019*). The labor and time-intensive complicated preprocessing and feature construction of traditional computer vision models pose severe challenges.

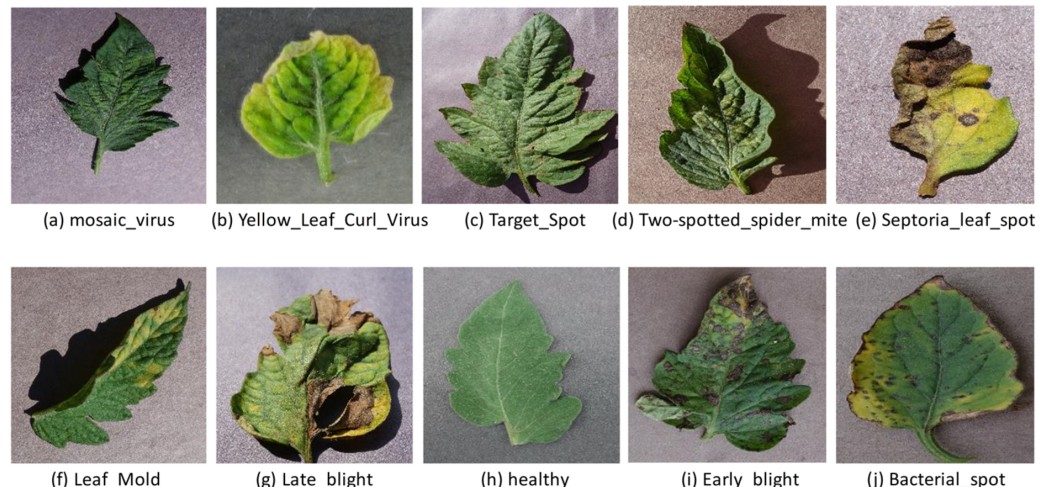

(a) mosaic_virus (b) Yellow_Leaf_Curl_Virus (c) Target_Spot (d) Two-spotted_spider_mite (e) Septoria_leaf_spot

(f) Leaf_Mold (g) Late_blight (h) healthy (i) Early_blight (j) Bacterial_spot

**Figure 1** **Illustrative images of the most prevalent illnesses affecting tomato leaves. Several diseases can cause spots and blemishes on leaves.** The image obtained from the plant village (tomato leaf disease dataset) images pixel size is 256 × 256 (*Hughes & Salath, 2015*): mosaic virus (A), yellow leaf curl virus (B), target spot (C), two-spotted spider mite (D), septoria leaf spot (E), leaf mold (F), late blight (H), healthy (I), early blight (J), bacterial spot (https://data.mendeley.com/datasets/tywbtsjrjv/l).

Furthermore, the precision of feature extraction procedures and the learning algorithm used to assess the effectiveness for leaf disease detection (*Saleem, Potgieter & Arif, 2019*). Due to advances in computer power, storage capacity, and accessibility of big data sets, deep learning technology has recently been applied to plant disease detection, a field gaining popularity in disease diagnosis due to advancements in computer power, storage capacity, and the availability of big data sets (*Rehman et al., 2024*). Convolutional neural networks (CNN) are one of the most popular methods for object identification, semantic segmentation, and image classification in the deep learning environment (*Widiyanto, Wardani & Pranata, 2021*; *Shahzad et al., 2023*; *Yazdan et al., 2022*). CNN-based deep learning models effectively extract features and learn non-linear correlations in the given data (*Liu et al., 2024*).

In this research, we develop a T-Net model to identify tomato foliar disease. Our models, which used deep learning, have multiple convolution layers, with each convolution layer selected with a schematization function and best normalization to improve energy efficiency and future extraction. We guarantee proportionality and accuracy in disease identification by incorporating the latest and most advanced techniques like dropout regression and submit activation. Our T-Net model is designed to examine tomato leaf images and extract complex patterns and features of various diseases. In addition, we use regularisation and data augmentation techniques to improve the training, which reduces overfitting and makes the modern moral generation of the MARA. The model builds on deep learning, significantly advancing tomato disease detection and control. We have combined state-of-the-art architecture with experimental setup and additional techniques to create a robot and flexible system to accurately identify tomato leaf disease interplay between the reprocessing pipeline and our model design of old agriculture environments.

Contributions of our study are as follows:

- Develop a lightweight T-Net model based on CNN to identify and classify tomato leaves and compare results to existing related research.
- Develop a comprehensive pre-processing pipeline to enhance the ability of the proposed T-Net model and existing DL models to generalize and improve the accuracy of tomato disease classification.
- Highlight the empirical effectiveness and computational efficiency of the proposed T-Net model over the existing CNN based DL models.
- Furthermore, a comprehensive analysis is given to highlight the findings of proposed T-Net model to address the challenges of changing agricultural environments by developing effective methods for identifying and treating tomato diseases.

The rest of this paper is organized as follows: 'Introduction' presents a general introduction to Tomatoes and their disease. 'Related Work' is Related Works on Tomatoes Classifications, and 'Proposed Model' is a detailed methodology of the system that explains how our system works. 'Performance Analysis' discusses performance analysis and experimental setup. 'Result and Discussion' is about results, compression with past results, confusion matrix, and overall study concludes in 'Conclusion'.

## RELATED WORK

With the rapid evolution of artificial intelligence (AI), deep learning (DL) has achieved significant strides in tackling computer vision challenges. Numerous classical computational deep learning models have been meticulously refined and expanded upon over the past twenty years. Research literature has documented the effectiveness of various models in discerning and diagnosing diverse plant diseases. Recently, a pioneering architectural innovation emerged, combining deep learning techniques with Squeeze and Excitation (SE) modules tailored explicitly for data analysis. The identification of plant diseases has long been researched. Plant disease identification has been studied for an extended period. Many methods have been developed for detecting tomato diseases, including color-focused algorithms (*Lubis et al., 2023*), texture (*Hlaing & Zaw, 2018*), or form of tomato leaves (*Kaur, Pandey & Goel, 2019*). Support vector machine (SVMs), decision trees (DTs), or neural network (NN) based classifiers were the main focus of early plant leaf disease detection. Visual spectrum images from professional cameras are used for disease detection in tomato leaves. Under laboratory circumstances, the acquired images were processed using clustering and step-wise multiple linear regression methods. Notably, the sample populations for the two studies were 180 samples for the second experiment and ranged from 22 to 47 for the first approach.

CNNs have quickly emerged as one of the most popular techniques for plant disease identification (*Lakshmanarao, Babu & Kiran, 2021*). Certain studies have concentrated on finding higher-quality characteristics by removing the constraints caused by homogeneity and illumination in complicated environmental scenarios. Several writers have created real-time models to speed up identifying plant diseases. Models developed by other authors have helped in the early diagnosis of plant diseases (*Liu & Wang, 2020*). *Mim et al. (2019)* utilized

images of tomato leaves to identify various illnesses. The authors develop a classification model using CNN and artificial intelligence (AI) algorithms, achieving a 96.55% accuracy rate in identifying five disorders. In *Huang et al. (2022)*, the authors introduced a self-paced learning approach to obtain more sparse classification and interoperable results. Similarly, in *Yin et al. (2024)*, the authors suggested a hybrid classification strategy combining convolutions and transformers to extract and learn features from image data efficiently. In addition, in *Zhang et al. (2024a)* and *Yang et al. (2024)*, classification approaches based on a few-shot learning and multi-path guidance networks are developed to enhance the accuracy of visual classification tasks. However, a few studies have assessed the effectiveness of deep neural network models when used to detect tomato leaf diseases. For example, in *Elfatimi, Eryiğit & Shehu (2024)*, the authors compared the performance of the Le-Net, VGG16, ResNet, and Xception models in classifying nine different types of diseases. The study solved the identical problem presented in *Gangadevi et al. (2024)* using the AlexNet, GoogleNet, and LeNet models; the accuracy results ranged from 94% to 95%. Using a dataset of 300 images, we utilized a tree classification model and segmentation to identify and categorize six distinct forms of tomato leaf disease (*Paul et al., 2023*). A method with a 93.75% accuracy rate for identifying and categorizing plant leaf disease has been suggested by *Salih (2020)*. Plant leaf disease is more accurately detected and classified thanks to image processing technologies and classification algorithms (*Imanulloh, Muslikh & Setiadi, 2023*). Sample data is gathered using a smartphone camera with 8 megapixels, and it is split into 50% categories for healthy and 50% categories for unhealthy users. Three steps comprise the image processing: enhancing contrast, segmenting the image, and extracting features. Two network architectures are examined, and classification tasks are carried out using an artificial neural network that uses a multi-layer feed-forward neural network. The mission distinguishes between the healthy and sick portions of the plant blade image; it cannot identify the type of illness. The authors employed color space analysis, color time, histogram, and color coherence to identify leaf illnesses and obtain 87.2% classification accuracy (*Sabrol & Kumar, 2016*). In addition, in *Huang, Shu & Liang (2024)*, the authors employed a learning algorithm using multi-omics data to capture biological processes to enhance diagnostic performance.

The circumstances of a tomato plant have been determined using a basic CNN model that contains eight hidden layers. Compared to other traditional models, the suggested strategies in *Kaur & Gautam (2021)* produced optimal results. The image processing system recognizes and categorizes tomato plant illnesses using deep learning techniques (*Goel & Nagpal, 2023*). The author implemented a whole system using CNN and the segmentation approach. To achieve higher accuracy, we have made modifications to the CNN model. In addition, *Xu, Li & Chen (2022)*; *Chen et al. (2023)*, the authors used pre-trained DL models to remove specular highlights from gray-scale images to improve the classification accuracy. By analyzing various spectral responses of leaf blade fractions, hyper-spectral images are used to diagnose rice leaf illnesses, including sheath blight (ShB) leaf diseases (*Kaur et al., 2024*). The author used CNN, segmentation, and image processing to categorize leaf illness. This study will classify and detect tomato illnesses affecting greenhouses and outdoor plants. Using the image from the sensor, the author employed deep learning and a robot to

detect plant illnesses in real-time. Various illness samples have generated a spectral library (*Sharma & Jindal, 2023*).

The literature review demonstrates how methods for identifying tomato leaf diseases have evolved, moving from conventional classifiers that relied on color, texture, and leaf shape to more sophisticated approaches like convolutional neural networks (CNNs). Current research shows that CNN-based models may achieve high accuracy rates (*e.g.*, 96.55% accuracy in diagnosing different tomato leaf diseases). The performance levels of various CNN designs, such as VGG16, ResNet, and AlexNet, vary; some models may achieve up to 99.25% accuracy. Developments in image processing technology and classification algorithms also allow more precise plant disease identification and classification. Improved CNN models and hyperspectral imaging significantly improve disease diagnosis, demonstrating the ongoing innovation and progress in plant disease detection techniques.

Moreover, improving a network's scope and depth will raise the number of parameters and increase error rates, which is the cause of overfitting in the results. Additionally, the computational cost of such a network develops with its complexity, providing problems for real-time implementation in agricultural challenges. A thoughtful distribution of computer resources is necessary due to the limitations of computational resources. Given these factors, this study introduces a lightweight classification model for tomato leaf identification. The suggested model is more appropriate for agricultural applications since it successfully addresses problems with lightweight network designs, reduces the number of training parameters, and improves training stability.

## PROPOSED MODEL

Deep learning techniques have substantially influenced image processing, plant disease detection, and categorization. As part of our research, deep learning techniques were used to revolutionize the field of image processing, particularly in detecting tomato plant leaf diseases (*Radovanovic & Dukanovic, 2020*). A new DL model, T-Net, was introduced and tailored to identify and categorize tomato diseases. The dataset was collected from the publicly available "PlantVillage" database (*Hughes & Salath, 2015*; *Pandian & G, 2019*). The total images of 16,569 are used in this research before augmentation. We classified diseases into 10 classes and fed tomato images into our T-Net model to evaluate their effectiveness. We also designed a T-Net architecture with Convn layers. We reorganized the dataset to ensure balanced training data, adjusting the number of images per disease class to between 1,500 and we employed traditional image data augmentation techniques to address imbalances in the dataset. For instance, we increased the number of images for classes like leaf mold and tomato mosaic virus by augmenting existing data through contrast, brightness, and horizontal flipping adjustments. Conversely, for diseases like tomato yellow leaf curl virus (TYLCV), where the dataset was overrepresented, we reduced the number of images to achieve balance and prevent bias in the classification network. The details of the restructure data are in Table 1. We aimed to utilize the capabilities of DL to enhance crop management and agriculture practices through the development of such a

**Table 1** Determine the number of images linked to each disease class and specify the scientific names of tomato leaf diseases.

| Disease name | Disease scientific name | Image number |
|---|---|---|
| Bacterial spot | *Xanthomonas campestris pv.Vesicatoria* | 2,127 |
| Early blight | *Alternaria solani* | 1,000 |
| Late blight | *Phytophthora infestans* | 1,909 |
| Leaf mold | *Fulvia fulva* | 952 |
| Septoria leaf spot (SLS) | *Septoria lycopersici* | 1,771 |
| Two-spotted spider mite (TSSM) | *Tetranychus urticae* | 1,676 |
| Target spot | *Corynespora cassiicola* | 1,404 |
| Tomato mosaic virus (TMV) | *Tomato mosaic virus* | 373 |
| Tomato yellow leaf curl virus (TYLCV) | *Begomovirus (Fam. Geminiviridae)* | 5,357 |

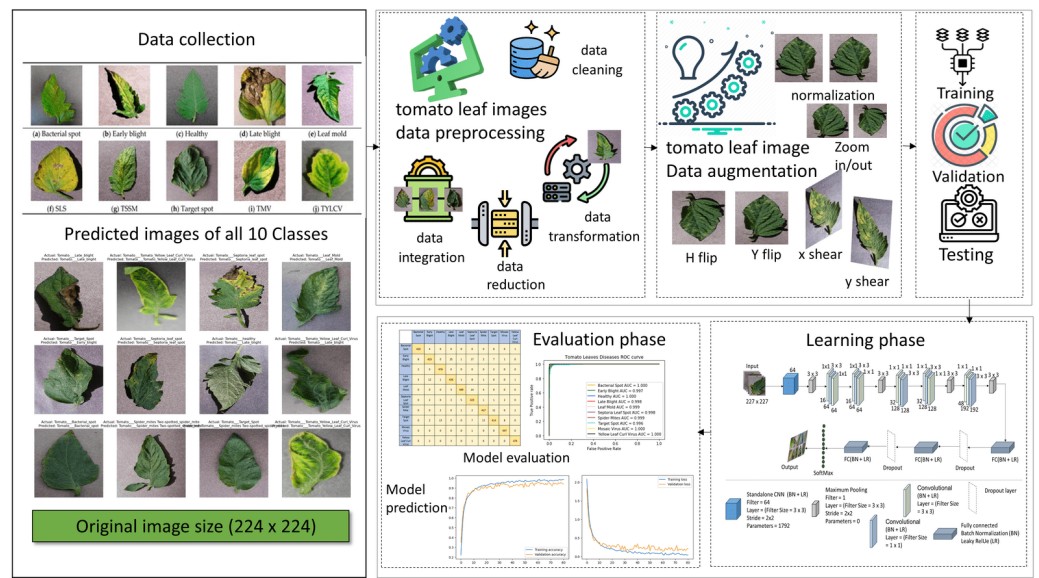

**Figure 2** The proposed method and the steps involved and the image used in the figure collected from the plant village tomato leaf dataset, *Pandian & G (2019)* for the plant leaf images we access data **Open source dataset link**. For the icon using in figure, we access through data preprocessing, data cleaning, data transformation, data reduction, data intergration, data augmentation, training, validation, testing.

robust framework for accurately detecting and categorizing tomato diseases. We provided a detailed, step-by-step explanation of our method in Fig. 2. It outlines each stage clearly and is easy to understand.

In preprocessing, we divide by 255, and the image's pixel values are scaled to fall between 0 and 1. Shearing modifications are applied to the photos at random. The zoom range of the image is subjected to arbitrary zooming adjustments, flipping images horizontally at random. The CNN model uses two bases: the advanced model and another that uses a VGG16 base for transfer learning. Convolutional layers, batch normalization layers, flattening layers, max-pooling layers, dropout layers, activation functions (ReLU), and

dense (ultimately linked) layers are the components of these models. Spatial dimensions and prevent overfitting, the model topologies include multiple convolutional layers with increasing filter sizes and depths, followed by max-pooling layers. Dropout layers randomly remove units from the model during training to regularize and avoid overfitting. Model compilation and training The binary cross entropy loss function and the Adam optimizer are used to compile the models. The data generators (train_generator and val_generator) are sent into the fit_generator function, which trains the model. A predetermined number of epochs and the model's performance are tracked using the validation data. Data preparation is essential to train the CNN model using tomato photos efficiently. Normalization and resizing are the two primary components of the data preparation operations. First, resizing guarantees that every image is standardized to a single size, which is necessary to make the CNN model's input layer compatible. All photos are resized to a consistent dimension (*e.g.,* 256 × 256 pixels) to reduce variances in image size with 224 × 224 and improve the model's ability to learn features across samples. Second, using the normalizing technique, the images' pixel values are scaled to a joint range, usually between 0 and 1. By preventing significant pixel value differences from controlling the optimization process, this normalizing phase is essential for improving the convergence and stability of the training process. Furthermore, data augmentation methods like horizontal flipping, shearing, and zooming are frequently used to vary the training sample further, improving the model's capacity to generalize well to new data. The input data provided to the CNN model must be appropriately prepared, standardized, and enhanced by these preprocessing steps. It offers an adequate basis for accurate tomato disease classification and robust model training.

## Feature extraction

In our proposed study, the ALVIN methodology enhances tomato plant disease detection by employing active learning with a lightweight T-Net model. Initially trained on small, labelled subsets, the model uses uncertainty sampling to identify complex patterns in a large pool of unlabelled images. Convolutional kernels are refined based on these uncertainties to improve feature extraction, capturing critical patterns like texture and colour variations. Expert feedback is incorporated to label challenging images, and the model is trained iteratively. This process reduces manual labeling efforts, enhances feature extraction, and improves the model's accuracy and generalization while maintaining low computational cost. The method also allows the model to adapt effectively to novel diseases and ensures efficient performance without excessive computational overhead. By refining kernels based on uncertain data, it strengthens the model's ability to extract meaningful features from complex image inputs. Mathematically, we can represent the convolutional operation as explained in Eqs. (1), (2) and (3):

$$(I * K)(x,y) = \sum_i \sum_j I(i,j) K(x-i, y-j). \tag{1}$$

Here:

- $I$ is the input image.
- $K$ is the convolutional kernel or filter.

- $(x, y)$ are the coordinates of the output feature map.

These convolutional filters essentially slide over the input image, performing calculations at each position to extract specific features. It's like looking through different windows to see what's inside.

A popular choice is the rectified linear unit (ReLU), which is a simple yet effective function:

$$f(x) = \max(0, x). \tag{2}$$

This function replaces all negative values with zero, while leaving positive values unchanged, similar to flipping a switch—if there's no signal, it turns off; if there's a signal, it passes through. Another technique used in Alvin's methodology is max pooling, which reduces the spatial dimensions of the data by selecting the maximum value from a region, effectively retaining the most important features

$$\max \text{ pooling}(x, y) = \max\_i, jI(x + i, y + j). \tag{3}$$

The pooling layer reduces computational complexity, simplifies the subsequent layers, and retains only the most essential information. This allows the model to focus on key features and patterns in the input images. Using specialized tools, like glasses for clarity, a tuning fork for precision, and a magnifying glass for detail, the model is further empowered to accurately identify tomato disease failures. These combined techniques enhance the model's ability to recognize critical distinguishing features within the images.

## Data augmentation

Data augmentation is an important preprocessing method for enhancing the diversity and strength of the training dataset in image classification problems. It reduces overfitting and improves the model's capacity for generalization by artificially increasing the dataset with modified copies of the original images. The Keras library's ImageDataGenerator class makes data augmentation easier by transforming input images in several ways. This code illustrates augmentation methods that help build a more resilient model, including shearing, zooming, rescaling, and horizontal flipping.

- Rescaling: By dividing each pixel value by 224, the image's pixel values are rescaled to fall between (0, 1). Since this normalization, optimization is more reliable and effective since all pixel values are kept within a constant numerical range.
- Shearing: Shearing transformations move pixels along the horizontal or vertical axes to randomly deform the images. Introducing heterogeneity in the orientation of items inside the images helps strengthen the model's resistance to various viewpoints and orientations.
- Zooming: Zooming changes arbitrarily enlarge or reduce the size of certain image regions. The model can learn characteristics at different degrees of detail thanks to this scale variation, which enhances its capacity to identify things at varied sizes and distances from the camera.
- Horizontal flipping: This technique randomly reflects the images along the vertical axis. By feeding the model photos with objects oriented both left-to-right and right-to-left,

this modification contributes to the dataset's diversity. The dataset is essentially increased by using these augmentation strategies, exposing the model to various variances and deformations that may occur in real-world situations. This leads to better performance and generalization on unobserved data as the trained model becomes more robust against noise, changes in illumination, and varied object orientations.

## Proposed T-Net model

This section presents step-by-step ALVIN methodology for developing an enhanced lightweight T-Net model based on CNN architecture for tomato plant leaf disease detection.

### *Proposed Alvin methodology*

ALVIN methodology is known as Active Learning through Verification of Interesting Knowledge. It uses active learning to minimize the need for large amounts of labeled data. In our proposed study, the ALVIN methodology consists of several steps:

**Train a lightweight T-Net model**: A lightweight T-Net model based on CNN architecture is developed and trained on small, labeled subsets of tomato leaf images covering different diseases. The lightweight trained T-Net model specifically developed for tomato plant leaf disease detection, focusing on low computational cost while maintaining accuracy.

**Apply active learning**: After initial training, apply the T-Net model to a large pool of unlabeled images of tomato plant leaves. Use active learning techniques and uncertainty sampling to identify patterns of the tomato plant disease images where the model is uncertain or likely to make errors. Based on the identified uncertainty, our proposed T-Net can adjust the convolutional kernels to better capture the relevant features, improving the feature extraction process. In contrast, traditional CNN uses convolutional kernels (or filters) to extract features from input data by performing convolutions at various layers.

**Refinement of convolutional kernels**: Once the input images' uncertain features are identified, convolutional kernels in the T-Net model are refined to focus on these challenging areas. Therefore, kernels in earlier layers are modified to capture better subtle texture differences or color variations in diseased *vs.* healthy tomato leaves, or deeper layers may be adjusted to focus on more complex patterns like lesion shape or size.

**Domain expert knowledge**: After identifying the most uncertain or informative images, the model queries a domain expert (such as lant pathologist) to label these images. This ensures that the model receives accurate labels for the most challenging images, improving its performance in these cases.

**Re-training of the model**: The newly labeled tomato plant disease images are added to the training set, and the T-Net model is refined and retrained. This step helps the T-Net model refine its feature extraction by modifying convolutional kernels, improving its ability to distinguish between different diseases, especially for complex or ambiguous cases.

**Iterative learning**: This process of identifying uncertain tomato plant disease images, querying for labels, refining convolutional kernels, and retraining the model is repeated in cycles. Over a few cycles, the T-Net model equipped with optimized convolutional kernels achieves high accuracy in detecting tomato plant diseases. It also becomes more accurate

and requires fewer queries from the domain expert, making it more efficient and capable of generalizing well on new, unseen tomato plant leaf images.

Furthermore, ALVIN methodology focuses only on the most uncertain or informative images, which minimizes the need for extensive manual labeling, reducing costs and time. In addition, the performance of the proposed T-Net model is improved because it learns from its mistakes or areas of uncertainty, resulting in better generalization and more accurate disease detection with fewer labeled examples. The ALVIN methodology also enhances the adaptability of our proposed T-Net model by enabling it to handle novel, previously unseen diseases. This is achieved through an intelligent querying of domain expert knowledge, allowing the model to evolve and respond to new disease types effectively and continuously. The T-Net architecture is developed to be lightweight, so combining it with ALVIN's method ensures that kernel adaptation and active learning do not add excessive computational overhead.

### Detailed overview of the proposed T-Net

The T-Net model is based on deep learning techniques for image classification applications. This architecture is divided into layers, which extract hierarchical information from the input images and predict output based on the layers. Convolutional layers comprise the T-Net architecture's framework and identify spatial patterns and characteristics in the input images. This particular design stacks many convolutional layers. Each convolutional layer convolves the input image with tiny receptive fields to create feature maps using a series of learnable filters. These feature maps draw attention to significant visual patterns in the images, such as edges, textures, and forms. Various activation functions (ReLU) and convolutional layers extract more sophisticated input image features (*e.g.*, 32, 64, and 128).

Max-pooling layers are used to downsample the feature maps after each convolutional layer, keeping the most pertinent data while lowering the feature maps' spatial dimensions. By dividing the feature maps into smaller sections and keeping just the maximum value inside each zone, max-pooling performs this downsampling. As a result of this process, the model is more robust in its ability to resist spatial translations and changes in object placement. Figure 3 shows the whole summary of the model in our proposed system T-Net model with its multiple layers. The architecture consists of flattening and fully connected dense layers as well as convolutional and max-pooling layers. The dense layers operate as the classifier by executing high-level reasoning based on the retrieved features, while the flattening layers turn 2D feature maps into 1D vectors. The given code further processes the flattened features and produces predictions using activation functions (ReLU) and dense layers divided into different numbers (128, 64, *etc.*) The T-Net model incorporates the "Fire module" and "squeeze layer" terminology, which are inspired by the SqueezeNet architecture developed by *Iandola et al. (2016)*. This architecture enhances efficiency by using squeeze layers that consist of $1 \times 1$ convolutions followed by expand layers that mix $1 \times 1$ and $3 \times 3$ convolutions. As described in SqueezeNet, the Fire module effectively reduces the number of parameters while maintaining performance, making it an optimal choice for lightweight models.

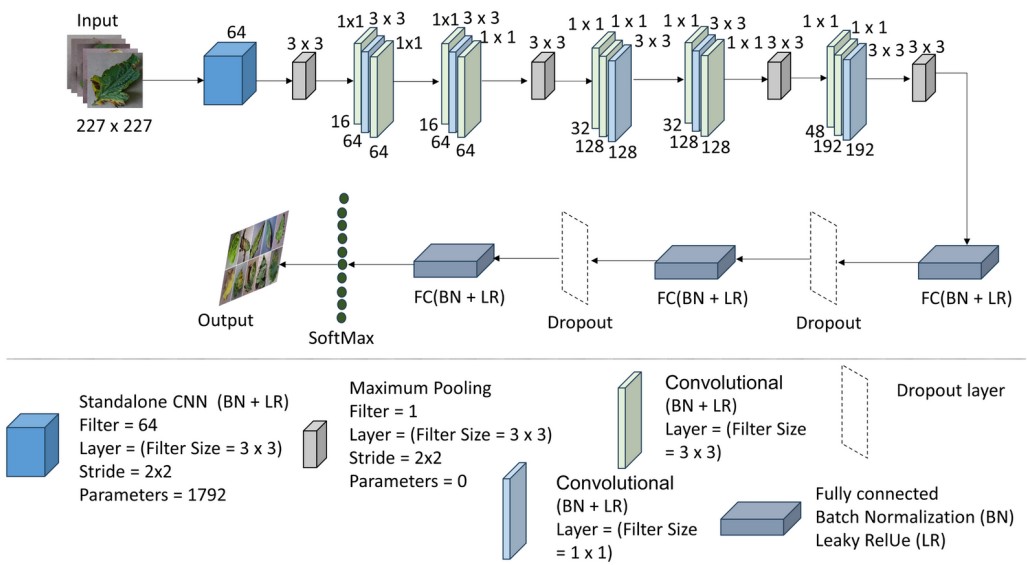

**Figure 3** The architecture of the T-Net model (*Iandola et al., 2016*; *Pandian & G, 2019*; https://data.mendeley.com/datasets/tywbtsjrjv/1.

Unlike ReLU, Leaky ReLU allows non-zero outputs for negative inputs, thus maintaining the flow of information and enhancing the model's classification performance. Table 2 shows the architecture of the T-Net module, a pivotal component enhancing deep learning-based object detection systems.

The output from the standalone convolution (first convolution layer) flows into the Fire 1 module, explicitly targeting its initial squeeze layer (the second convolution layer with a 1 × 1 filter), following normalization, activation, and pooling. Subsequently, the output of the first squeeze layer progresses to the third convolution layer (the first expand layer), implementing 64 filters of size 3 × 3 with a padding of 1 pixel. This expanded layer's output then advances to the fourth convolution layer (expand layer), which utilizes 64 filters of size 1 × 1. This process iterates, with each Fire module's output forwarded to the following Fire module's initial convolution layer (squeeze layer). Finally, the output of the fifth Fire module is directed to the first fully connected dense (FCD) layer, which converts the two-dimensional feature map extracted by the convolution layers into a one-dimensional feature vector.

## PERFORMANCE ANALYSIS

This research evaluates the effectiveness of the proposed T-Net across 10 distinct tomato leaf disease classes: BS, LM, SLS, TMV, and TYLCV. Data augmentation is applied explicitly to the 10 classes, while the data for the TYLCV class is restructured to improve model performance. Evaluating the T-Net classifier's effectiveness involves a comprehensive analysis across eight performance metrics, including parameters count, accuracy, error rate, precision, recall, sensitivity, specificity, and F1-score. Comparative analysis is conducted against seven contemporary transfer learning models. The subsequent section provides

**Table 2  Overview of the architecture of the T-NetFire Module, a pivotal component enhancing deep learning-based object detection systems.**

| No | Operation | Layer | Filters | F size | Padding | Stride | Parameters |
|----|-----------|-------|---------|--------|---------|--------|------------|
| 1 | | Input | | | | | 0 |
| 2 | Standalone CNN | Convolutional (BN + LR) | 64 | 3 × 3 | — | 2 × 2 | 1,792 |
| 3 | Pooling layer | MaximumPooling | 1 | 3 × 3 | — | 2 × 2 | 0 |
| | | Convolutional (BN + LR) | 16 | 1×1 | — | — | 1,040 |
| 4 | Fire one | Convolutional (BN + LR) | 64 | 3 × 3 | [1 1 1 1] | — | 9,280 |
| | | Convolutional (BN + LR) | 64 | 1×1 | — | — | 4,160 |
| | | Convolutional (BN + LR) | 16 | 1×1 | — | — | 1,040 |
| 5 | Fire two | Convolutional (BN + LR) | 64 | 3 × 3 | [1 1 1 1] | — | 9,280 |
| | | Convolutional (BN + LR) | 64 | 1×1 | — | — | 4,160 |
| 6 | Pooling layer | MaximumPooling | | 3 × 3 | [0 1 0 1] | 2 × 2 | 0 |
| | | Convolutional (BN + LR) | 32 | 1×1 | — | — | 2,080 |
| 7 | Faire three | Convolutional BN + LR | 128 | 1×1 | — | — | 4,224 |
| | | Convolutional (BN + LR) | 128 | 3 × 3 | [1 1 1 1] | — | 147,584 |
| | | Convolutional (BN + LR) | 32 | 1×1 | — | — | 16,512 |
| 8 | Fire four | Convolutional (BN + LR) | 128 | 3 × 3 | [1 1 1 1] | — | 36,992 |
| | | Convolutional (BN + LR) | 128 | 1×1 | — | — | 16,512 |
| 9 | Pooling layer | MaximumPooling | | 3 × 3 | [0 1 0 1] | 2 × 2 | 0 |
| | | Convolutional (BN + LR) | 48 | 1×1 | — | — | 6,192 |
| 10 | Fire five | Convolutional (BN + LR) | 192 | 1×1 | — | — | 9,408 |
| | | Convolutional(BN + LR) | 192 | 3 × 3 | 1×1 | — | 331,968 |
| 11 | | FC + BN + LR + Dropout | | | | | |
| 12 | | FC + BN + LR + Dropout | | | | | |
| 13 | | FC + Soft max + classification | | | | | |

detailed descriptions of the experimental setup and dataset characteristics, offering a thorough understanding of the analysis conducted.

## Experimental setup

The experiment was conducted on a 64-bit operating system, an x64-based server, housing an Intel(R) Core(TM) i7-8700 CPU @ 3.20 GHz, which can turbo boost up to 3.19 GHz. This CPU is known for its high performance, especially in tasks requiring significant computational power. Moreover, to further enhance the server's capabilities, it was augmented with an NVIDIA GeForce GTX GPU, which boasts 6 GB of GPU memory. The GPU operates at frequencies up to 1060 MHz, making it suitable for daily parallel processing tasks in deep learning applications. Its single-precision performance, rated at six TE-LOPS, underscores its ability to handle complex computations efficiently. To utilize powerful computational resources, such as GeForce GTX GPU, to primarily reduce the training time of the proposed lightweight T-Net architecture. It also provided us with real-time feedback to reduce the training time, which was invaluable for refining the proposed lightweight T-Net architecture. The secondary aim was to handle real and augmented image data more efficiently than traditional CPU-based machines. The proposed model was implemented using TensorFlow, a widely adopted deep-learning framework known for its flexibility and

**Table 3  Configuration of the proposed model system's environment involves the setup process.**

| Configuration resource | Value details |
| --- | --- |
| Device name | DESKTOP-KIQGQ3F |
| CPU | Intel(R) Core(TM) i7-8700 CPU @ 3.20 GHz 3.19 GHz |
| RAM | 32.0 GB |
| GPU | NVIDIA GeForce GTX 1060 6Gb |
| Software | Jupyter Notebook |
| Language | Python |
| Operating system | 64-bit operating system, x64-based processor |

scalability. In Table 3, the hardware and software configurations utilized in the study are detailed, providing a comprehensive overview of the experimental setup.

## Performance measure

After preprocessing, the dataset comprising 10,000 images of tomato diseases was prepared for experimentation. The dataset was divided into training and testing subsets to facilitate model training and evaluation, with proportions of 80% and 20%, respectively. Deep learning models automatically extracted disease features *via* convolution operations, eliminating the need for manual feature extraction. The classification performance evaluation varied between deep learning approaches to assess performance; the proposed network is bench-marked against well-known CNN architectures, including the transfer learning models VGG-16, Inception V3, and AlexNet. The standard evaluation metric for image classification, average accuracy, is employed. This evaluation employs distinct metrics such as accuracy, precision, recall, and F1 score. These metrics were calculated using Eqs. (4), (5), (6) and (7).

$$Accuracy = \frac{TP+TN}{TP+TN+FP+FN} \tag{4}$$

$$Precision(P) = \frac{TP}{TP+FP} \tag{5}$$

$$Recall = \frac{TP}{TP+FN} \tag{6}$$

$$F1Score = \frac{2PR}{P+R}. \tag{7}$$

## Parameter setting

In this experiment, we chose a batch size of 32 to make our process more efficient and address the issue of insufficient data. When we use larger batch sizes, our classification accuracy decreases because the learning rate decreases. We kept all the model settings at their default values. During training, we went through the data in batches, updating the

**Table 4  Hyperparameters are configuration settings used to tune the behavior of the learning algorithm.**

| Hyperparameter | Values |
|---|---|
| Dropout rate | 0.5 |
| Batch size | 32 |
| Activation function | ReLU |
| Learning rate | 0.01 |
| Epoch | 80 |
| optimizer | Adam |

neural network's weights as we went along throughout 80 epochs. We wanted our models to get as good as possible before they started to over-fit, so we stopped at 80 epochs. We used a fixed learning rate of 0.01 and ensured the models saved themselves automatically as they trained. We used the Adam optimizer because it's good at handling moving targets and situations where we don't have much information about the gradients. We used a particular loss function called sparse categorical cross-entropy for our classification tasks. We randomly dropped half of the connections between neurons during training to prevent our models from memorizing the training data too much (which is overfitting), which helped our models perform better. We tried many different combinations of settings while building our models to find the ones that worked the best. You can see the specific settings in Table 4.

## RESULT AND DISCUSSION

### Result

This section describes the performance of the proposed model and transfer learning-based models that have been utilized. Accuracy and loss graphs are used to understand the model behavior better. The ROC-AUC curve of all tomato leaf classes, a significant measure of the model's performance, has been demonstrated. Moreover, the proposed model was compared with other studies to determine reverence. The left graph in Fig. 4 illustrates how effectively the model learns from the training data and generalizes to unknown validation data by displaying the model's accuracy across epochs on both the training and validation datasets. The right graph displays the loss, showing the variation between the predicted and actual values during training and validation. Understanding the model's convergence, spotting possible over- or underfitting, and fine-tuning the model's parameters for better performance depend on these visual aids.

The graphical representation of our proposed model shows model accuracy, training, test, loss, and epoch of the model. The T-Net model's ability to categorize tomato images into categories corresponding to health or sickness is shown visually by the confusion matrix. The left graph in Fig. 5 displays the number of true positive, true negative, false positive, and false pessimistic predictions, offering valuable information on how well the model can categorize various cases. The confusion matrix visually represents these metrics, which helps evaluate the model's overall accuracy, precision, recall, and F1 score. By doing so, practitioners can identify biases or incorrect classifications and enhance reliability and
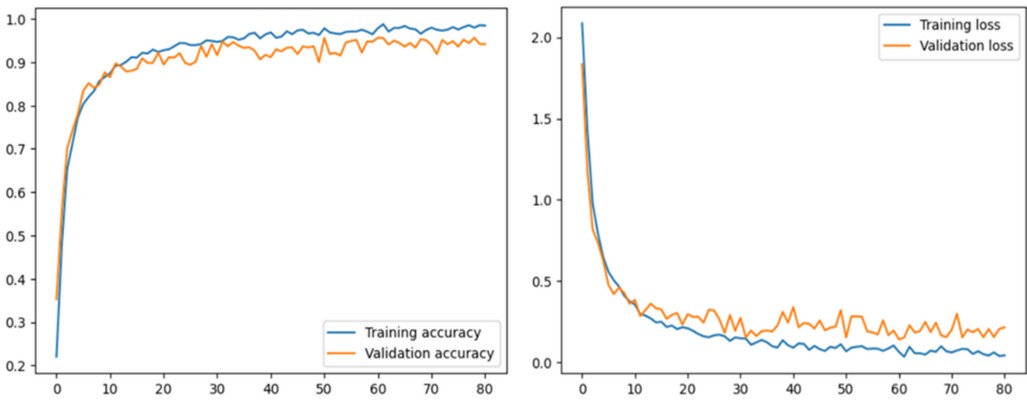

**Figure 4** Accuracy and loss graph of the proposed model.

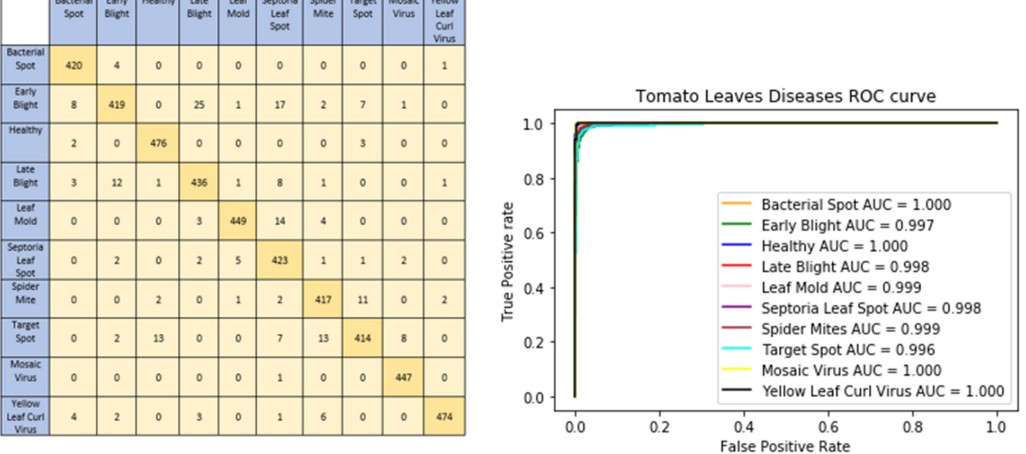

**Figure 5** Confusion matrix and ROC curve of the tomato leaf disease classes according to DL techniques.

performance. The right graph of Fig. 5 shows the ROC-AUC of the tomato leaf classes. The target spot of the tomato leaf has the lowest AUC score compared to the other classes. However, the bacterial spot performs better compared to other courses.

In addition, Table 5 outlines the performance metrics of a proposed model across various classes or categories within a dataset. The model demonstrates high precision and recall for classes like leaf mold, spider mites, two-spotted spider mites, tomato yellow leaf curl virus, and tomato mosaic virus, with F1-scores ranging from 0.94 to 0.98. Notably, the model achieves an exceptional F1-score of 0.97 for the healthy leaf class, with precision and recall values of 0.99 and 0.96, respectively, and an accuracy of 100%. However, classes like target spot and early blight exhibit slightly lower precision and recall scores, leading to comparatively lower F1-scores of 0.89 and 0.91, respectively. These metrics were calculated using macro-level averaging, guaranteeing that every class contributes equally to the final

**Table 5** Proposed model refers to evaluating the model's performance across individual classes or categories within the dataset, providing insights into its effectiveness for specific tasks or targets.

| Classes | Precision | Recall | F1-Score | Support |
|---|---|---|---|---|
| Bacterial_Spot | 0.96 | 0.95 | 0.95 | 100 |
| Early_Blight | 0.91 | 0.88 | 0.89 | 100 |
| Healthy | 0.99 | 0.96 | 0.97 | 100 |
| Late_Blight | 0.90 | 0.96 | 0.93 | 100 |
| Leaf_Mold | 0.98 | 0.93 | 0.95 | 100 |
| Septoria_Leaf_Spot | 0.90 | 0.92 | 0.91 | 100 |
| Spider_mintes T0-spotted_spider_mite | 0.98 | 0.91 | 0.94 | 100 |
| Target_Spot | 0.88 | 0.95 | 0.91 | 100 |
| Tomato_Yellow_Lef_Curl_Virus | 0.96 | 0.97 | 0.97 | 100 |
| Tomatoes_Mosiaic_Virus | 0.97 | 0.98 | 0.98 | 100 |

**Table 6** Comparison of the proposed model with based model.

| Model | Precision | Recall | F1 score | Accuracy |
|---|---|---|---|---|
| Inception V3 | 95% | 93% | 92% | 95% |
| AlexNet | 94% | 92% | 93% | 95% |
| VGG16 | 96% | 94% | 95% | 96% |
| **Proposed model** | **98.75%** | **98.78%** | **98.97%** | **97.90%** |

evaluation. They provide information on the model's general performance for particular tasks or targets within the dataset and how well it labels various classes.

Furthermore, Table 6 compares the empirical effectiveness of the proposed T-Net model with the existing DL models, such as Inception V3, AlexNet, and VGG16. Different metrics are used to evaluate the empirical effectiveness of the proposed T-Net model over the existing models, such as precision, recall, F1 score, and accuracy. The empirical effectiveness highlights that the proposed T-Net model yields better classification performance than the existing models. The proposed model significantly outperforms Inception V3, AlexNet, and VGG16. The proposed model demonstrates its ability to identify relevant instances and accurately minimize false predictions. Therefore, our proposed model with classification improvement over existing models helps farmers mitigate risk in advance to save large portions of the plants from being affected. In addition, our proposed model can lead to better plant disease detection compared to the existing models, which further helps farmers reduce plant losses and optimize resources.

## Discussion

The results demonstrate the performance of the T-Net model trained for image classification. First, the evaluation of model performance parameters, such as loss and accuracy, provides crucial clues about the predictive power of the models. High accuracy scores show robust classification abilities, and low loss values indicate the successful alignment of the predicted and accurate labels. These measures function as essential standards for assessing the effectiveness of the model.

More information about the learning dynamics of the models is obtained from the training and validation curves over epochs. When both curves smoothly converged, training was practical, and no overfitting occurred; nevertheless, when there were significant differences between the two, problems like under- or overfitting may have occurred. Furthermore, examining confusion matrices and viewing sample predictions aid in comprehending the categorization behavior of the models. Patterns in misclassifications may be found by analyzing right and wrong predictions, providing essential information for model improvement.

Understanding the relative performance of various model designs, preprocessing approaches, and regularization strategies requires comparative analysis. Adjusting hyperparameters such as batch size and learning rate may optimize model training, improving classification resilience and accuracy. Additionally, examining the models' scalability and generalizability to different datasets or domains clarifies their usefulness outside the current context. Future research paths and improvements are made possible by discussing the results. It emphasizes areas needing development, such as investigating more intricate designs, utilizing group approaches, or adding cutting-edge strategies like attention processes. Furthermore, resolving any restrictions or difficulties found during assessment encourages the creation of more efficient image classification models with broader applicability and improved performance.

The empirical findings of our proposed T-Net model was evaluated against existing models in Fig. 6 such as Inception V3, AlexNet, and VGG16, showing notable improvements in key performance metrics: accuracy, precision, recall, and F1 score. To substantiate the importance of these improvements, we conducted a paired sample $t$-test analysis, which confirmed that the enhancements were statistically significant, with $p$-values below 0.05 across all metrics. Specifically, the T-Net model's accuracy improvement of 3–5% over the baseline models was statistically significant ($p = 0.0083$), as was the increase in precision, particularly compared to AlexNet, which was higher by 5.05% ($p = 0.0229$). The T-Net model also demonstrated a substantial improvement in recall, outperforming AlexNet by 7.37% ($p = 0.0098$), which is especially significant for minimizing false negatives in practical applications. Additionally, the model achieved a 7.57% enhancement in F1 score over Inception V3 ($p = 0.0236$), indicating a balanced increase in both precision and recall. These findings validate the T-Net model's robustness and practical applicability for tomato disease classification. Even minor gains in accuracy can be crucial in real-world agricultural settings, reducing misclassifications and enabling farmers to manage plant diseases more effectively.

In addition, Table 7 compares the computational efficiency of the proposed T-Net model with the existing DL models. The computational efficiency analysis indicates that the proposed model requires fewer trainable parameters than the existing models. The proposed model reduces the number of parameters by approximately 37.17% compared to Inception V3. Similarly, our proposed model requires 96.03% fewer parameters than AlexNet, which indicates the efficiency of the proposed model to faster processing times. Besides Inception V3 and AlexNet, our proposed model reduces the number of parameters by approximately 98.25% compared to the VGG16. Hence, our proposed model achieves

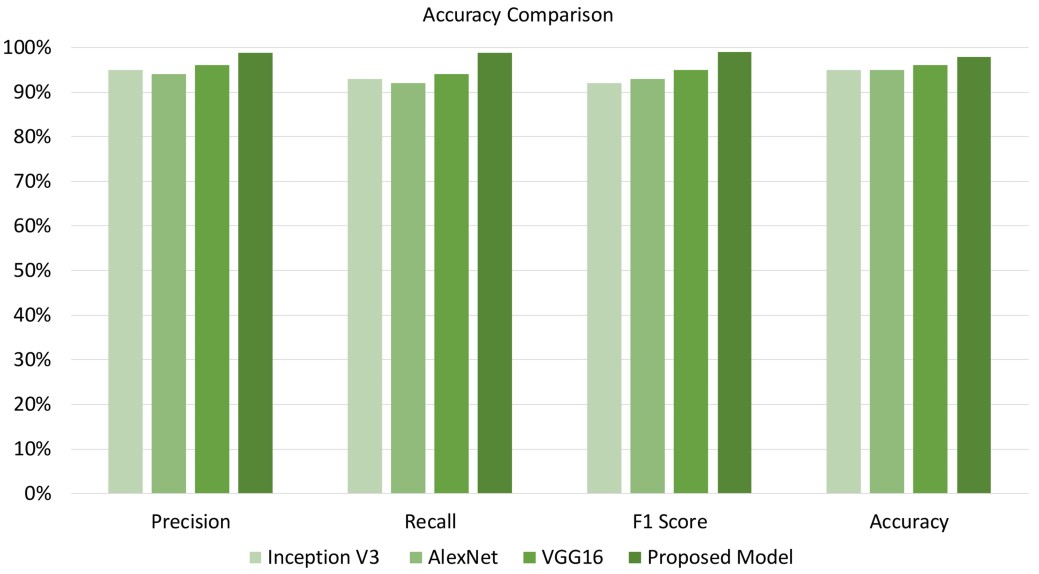

**Figure 6** Illustrating accuracy, precision, recall, and F1-score enables us to evaluate the model effectiveness on the best-performing approach.

**Table 7 Computational efficiency in terms of trainable parameters.**

| Model name | Number of total parameters |
| --- | --- |
| Inception V3 | 3,851,784 |
| AlexNet | 60,954,656 |
| VGG16 | 138,357,544 |
| Proposed model | 2,420,615 |

high accuracy and offers substantial improvements in computational efficiency compared to existing deep learning models. This makes it an efficient and impactful solution for applications in tomato agriculture.

Furthermore, to evaluate the model's efficacy in incorrectly identifying healthy and infected tomatoes, its performance metrics accuracy. The conversation may also cover identifying any difficulties discovered in developing the model, such as imbalanced data, overfitting, or underfitting, and suggest possible remedies or directions for more research. It would also be beneficial to investigate parallels with current approaches or earlier studies on categorizing tomato diseases to put the results in perspective and emphasize the improvements or originality of the T-Net model. In conclusion, the T-Net architecture in the given code uses convolutional, pooling, flattening, and dense layers to extract hierarchical features from input photos and provide predictions for binary classification tasks. This architecture is well-suited for various image identification and classification applications because it can automatically train discriminative features from raw pixel data. Table 8 compares multiple architectures used in research articles, each evaluated on different datasets for image classification tasks. In the study by LeafNet (*Tm et al., 2018*),

**Table 8  Comparisons of results with related studies indicate performance trends and validate findings in the broader research context using plant village dataset.**

| Ref | Year | Architecture | Accuracy | Limitations |
|---|---|---|---|---|
| *Tm et al. (2018)* | 2018 | LeafNet | 95% | Apply a single model without rewriting for deployment. |
| *Zhang et al. (2018)* | 2018 | MPC | 96% | Reduce both the number of classes and the size ofthe dataset |
| *Agarwal et al. (2020)* | 2020 | CNN | 91% | Decrease accuracy without implementing deployment changes. |
| *Chen et al. (2022)* | 2022 | AlexNet | 96% | Single model applied |
| *Ni et al. (2023)* | 2023 | ResNet | 96% | Model unable to adapt to varying conditions |
| *Chen et al. (2024)* | 2024 | CNN | 95% | Limitations exist in capturing localized feature representations. |
| Proposed model | 2024 | T-Net | 98.97% | May pose challenges due to resource constraints |

which utilized the Plant Village dataset consisting of 18,100 images across 10 classes, an accuracy of 95% was achieved. However, deployment posed challenges as the model was applied without any modifications. MPC (*Zhang et al., 2018*) achieved a slightly higher accuracy of 96% by reducing the dataset size and the number of classes to 5,000 images and 9 classes, respectively. MobileNet (*Elhassouny & Smarandache, 2019*) attained 90% accuracy on a self-collected dataset of 7,200 images with 10 classes, yet the smaller dataset size may limit its applicability. A CNN architecture (*Agarwal et al., 2020*) reached 91% accuracy on the Plant Village dataset but experienced a decrease in performance when deployed without modifications. SE-ResNet (*Ahmad et al., 2020*) achieved 96% accuracy on a smaller self-collected dataset of 4,600 images, indicating potential limitations in data availability. Both AlexNet (*Zhao et al., 2021*) and ResNet (*Ni et al., 2023*) achieved 96% accuracy on the Plant Village dataset but struggled with adaptability to varying conditions, raising concerns for real-world deployment. A CNN architecture in *Chen et al. (2024)* achieved 95% accuracy on the Plant Village dataset but was limited in capturing localized feature representations. The proposed model achieved the highest accuracy of 98.97% compared to other existing models.

### Theoretical implications
This research develops a unique T-Net architecture. First, the trained model's performance on previously unknown data shows how well it can apply newly acquired patterns to new situations. The model's ability to capture underlying patterns across several datasets through its consistent performance demonstrates the resilience of the selected architecture and learning strategy. This supports the idea that machine learning models may learn complex patterns in data and that these algorithms can reliably predict outcomes based on hypothetical cases. In addition, studying the learned feature representations offers essential insights into how the model processes incoming data. Convolutional neural networks

(T-Net) are examples of deep learning models that automatically learn hierarchical feature representations from unprocessed input. Gaining a better grasp of representations can help us better understand feature representation, learning in neural networks, and the discriminative characteristics the model has detected. The expressiveness of the model architecture is also significantly influenced by its complexity. More intricate models run the danger of overfitting to noise in the training set, even though they could perform better. Developing machine learning models that work effectively on unknown data requires balancing model complexity and generalization capacity.

### Challenges and Limitations

Developing the T-Net model for tomato disease classification faced several challenges. One major issue was the quality and availability of the dataset. Despite our efforts to collect a diverse range of tomato photos at different disease stages, we encountered problems like uneven class distributions and not enough samples for some diseases. To address this, we must emphasize the need for more annotated photos, as they are crucial for improving the model's accuracy. Careful data curation and data augmentation to diversify the dataset are also essential tasks. Another challenge was the complexity of accurately categorizing tomato diseases. Variations in disease appearance, background clutter, lighting, and image resolution made it hard for the T-Net model to identify diseases correctly. This made it difficult for the model to adapt to new or challenging situations in real agricultural settings.

A further limitation was the model's scalability and resource requirements. The deep T-Net design needs a lot of processing power and memory, especially with many layers and parameters. Training large T-Net models can be impractical for practitioners without high-performance computing equipment. Future studies should investigate more computationally efficient model designs, such as lightweight or trimmed versions of T-Net. Techniques like transfer learning, where we fine-tune pre-trained models on our dataset, can also reduce computational needs. Cloud-based platforms or federated learning can also allow distributed training, making it easier for those with limited local resources. These strategies can help overcome resource limitations while maintaining or improving the model's performance. The T-Net model's interpretability is a significant issue, especially in agricultural settings where stakeholders need clear insights into disease classifications. As a black-box model, T-Net can be hard to understand in terms of its decision-making process. To build trust and acceptance among agricultural practitioners, we must incorporate methods that improve interpretability or explore alternative models emphasizing transparency and explainability.

## CONCLUSION

In conclusion, our research represents an essential development in agricultural technology, particularly in identifying tomato foliar diseases. We have successfully identified and characterized the averseness of various tomato foliar diseases by applying deep learning techniques such as our new T-Net model. Every convolutional layer in our T-Net model has been evolved to extract sequential information from the input image, ensuring stability and accuracy in disease classification. Our solutions address real-world agricultural problems.

This comprehensively analyzes patterns in different data sets by incorporating explained prediction and augmentation techniques. Our approach prioritizes and uses practicality and provides solutions with practical knowledge for disease management. Further collaboration and synthetic improvements will allow farmers to maintain tomato leaf health despite changing synthetic environments. As a future work, our proposed lightweight T-Net architecture will be extended by employing explainability methods, such as model agnostic methods (*e.g.*, LIME and SHAP), gradient-based method (grad-CAM), *etc.*, to gain insights into how the proposed architecture processes the inputs to make classification decisions.

### Funding

This research was supported by the ''Regional Innovation Strategy (RIS)'' through the National Research Foundation of Korea (NRF) funded by the Ministry of Education (MOE)(2023RIS-009). The funders had no role in study design, data collection and analysis, decision to publish, or preparation of the manuscript.

### Grant Disclosures

The following grant information was disclosed by the authors:
''Regional Innovation Strategy (RIS)'' through the National Research Foundation of Korea (NRF) funded by the Ministry of Education (MOE): 2023RIS-009.

### Competing Interests

The authors declare there are no competing interests.

### Author Contributions

- Amreen Batool conceived and designed the experiments, performed the experiments, analyzed the data, performed the computation work, prepared figures and/or tables, authored or reviewed drafts of the article, and approved the final draft.
- Jisoo Kim performed the computation work, prepared figures and/or tables, authored or reviewed drafts of the article, and approved the final draft.
- Sang-Joon Lee performed the computation work, prepared figures and/or tables, authored or reviewed drafts of the article, and approved the final draft.
- Ji-Hyeok Yang performed the computation work, prepared figures and/or tables, authored or reviewed drafts of the article, and approved the final draft.
- Yung-Cheol Byun analyzed the data, prepared figures and/or tables, authored or reviewed drafts of the article, and approved the final draft.

### Data Availability

The dataset is available at Mendeley: J, ARUN PANDIAN; GOPAL, GEETHARAMANI (2019), ''Data for: Identification of Plant Leaf Diseases Using a 9-layer Deep Convolutional Neural Network'', Mendeley Data, V1, doi: 10.17632/tywbtsjrjv.1.

The training and validation data of tomato leaf disease is available at Kaggle: https://www.kaggle.com/datasets/amreenbatool/plant-leaf-disease-data.

The source code is available at GitHub and Zenodo:

- https://github.com/Amreen-source/Tomato-leaf-disease-detection-

- Amreen, B. (2024). Tomato-leaf-disease-detection [Data set]. Zenodo. https://doi.org/10.5281/zenodo.14020689.

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
