# Peer review of "An enhanced lightweight T-Net architecture based on convolutional neural network (CNN) for tomato plant leaf disease classification"

_PeerJ Computer Science, doi:10.7717/peerj-cs.2495_

## Round 0.1 · original submission · Major Revisions

The authors must revise the paper carefully.

Reviewer 1 ·

Basic reporting

1. The language requires revision for clarity and professionalism. Several areas suggest potential use of voice recognition software, leading to nonsensical phrases like "Our teenage model" (likely intended as "Our TNet model"). Sentences are often grammatically incorrect, lack proper punctuation (lines 86-91), and have random capitalization within sentences. It appears the manuscript was not proofread before submission.
2. Literature references are present, but the field background/context could be strengthened. Specifically, the abstract mentions a ranking of cultivated crops (tomatoes being third after potatoes and sweet potatoes), but the ranking source is unclear and irrelevant to the paper's focus on tomato leaf disease classification.

Experimental design

1. The research question (tomato leaf disease classification) appears relevant to the journal's scope. However, the introduction inadequately explains the novelty of the proposed model ("novel, deeper, and lightweight" in line 171-175). The paper could benefit from a clearer explanation of how it addresses a knowledge gap in the field.
2. The experimental setup description mentions a powerful server computer with GeForce GTX GPU, known for high performance according to the author. However, considering the model's supposed lightweight design, justification for such computational resources is lacking.
3. Section 1.1 on feature extraction mentions Alvin's methodology but only describes it as a filter/convolution kernel. Crucially, details on specific kernel parameters and how it differs from standard convolution kernels are missing.
4. The introduction includes ReLU activation function, which is appropriate. However, Sigmoid activation is also introduced but seemingly not used in the model. Removing unnecessary information would improve clarity.
5. The authors introduce the "Fire module" and "squeeze layer" terminology in line 291-299, likely referencing the SqueezeNet model. However, there's no prior introduction of this model or citation for its reference.
6. A discrepancy exists between the initial image count (16569) mentioned in line 182 and the final dataset size (10000) used for training and testing in section 2.2. This inconsistency needs clarification.

Validity of the findings

1. The claimed accuracy (98.97%) is impressive, but the comparison with other models is flawed. Baseline model accuracies are sourced from their original papers, which likely used different training and testing datasets and image numbers. This comparison lacks context and may be misleading due to underlying dataset differences.
2. The significance of the 3% improvement over existing models (around 95-96%) is not addressed. The paper should explain why this improvement is crucial and impactful for tomato agriculture.

Additional comments

The paper requires thorough revision to improve clarity, conciseness, and overall professionalism. The authors should address the specific points mentioned above to strengthen the research question, methodology description, and overall impact of their findings.

Reviewer 2 ·

Basic reporting

The paper presents a novel T-Net architecture based on CNNs for the classification of tomato plant leaf diseases, achieving a high accuracy. By integrating transfer learning from VGG-16, Inception V3, and AlexNet, the model outperforms existing methods in both accuracy and efficiency. Extensive experiments and comparisons validate its effectiveness, offering a dependable approach for diagnosing tomato diseases.
1. Are there any images/leaves classified into multiple diseases?
2. As the authors mentioned the limitation of resources to train a large model, the authors should provide future insights on how to address this limitation.
3. The authors should comment on the model’s interpretability in more detail. Running a SHAP analysis (or similar test on model interpretation) should strengthen the paper or make further suggestions.

Experimental design

The research question is well-defined, addressing the need for an efficient, accurate, and lightweight model for tomato leaf disease classification. The methodology is thoroughly described, enabling replication. The authors explain the architecture of the proposed T-Net model, detailing the data preprocessing steps and the experimental setup. They employed traditional image data augmentation techniques to balance the dataset and enhance the model's training process. The model combines elements of transfer learning using VGG-16, Inception V3, and AlexNet with a lightweight architecture to improve accuracy and efficiency. The experimental setup includes a comprehensive description of the hardware and software used, highlighting the computational resources required to train and evaluate the model.

Validity of the findings

The data used for training and testing the model is robust, with a substantial number of images representing various disease categories. The statistical methods employed to evaluate the model's performance, including accuracy, precision, recall, and F1 score, are appropriate and well-executed. The results are compelling, with the T-Net model achieving a high accuracy of 98.97%. The conclusions are well-supported by the results, demonstrating that the proposed model outperforms existing methods. The discussion effectively contextualizes the findings within the broader field, suggesting practical applications and future research directions. Overall, the study's findings are robust, statistically sound, and controlled, providing a significant contribution to the field of plant disease detection.

Additional comments

1. The citation style is confusing. The authors are recommended to check the journal requirements and use citation management software to clarify the manuscript.
2. There are grammar mistakes in the manuscript. Please check the manuscript carefully.
3. The figures need annotations. For example, there is no 1(a). For context, the authors should include a scale bar or mention the pixel size in the figure legend and state the picture's origin.
4. Something is missing in line 104, which makes the sentence hard to understand.
5. Lines 121-124, the sentences here need to be clarified.
6. Typo in Figure 6.

·

Basic reporting

No comments. The manuscript is overall nicely written.

Experimental design

The authors in this work is proposing a novel and interesting DL architecture, T-Net. It is claimed that this architecture enables better training ability and mitigates issues related to deep DL architecture. It has significantly fewer number of parameters compared to other state-of-the-art DL models while achieving better performance given the presented results. Data is well introduced (preprocessing steps, data imbalance) and ensures reproducibility. However, I do have a few comments.

1. For the metrics shown in figure 6, precision, recall, f1 score, are these macro or micro level metrics since it's multi-class labels?
2. To show whether T-Net is doing what is supposed to do, I'd encourage the authors to explore saliency based methods. This will give an idea if the model was looking for the right pattern for disease classification.

Validity of the findings

Just one nit picky, it seems the performance among various models in figure 6 is very close. Why don't the authors do cross validation or repeated training with data shuffling to get an idea on the variability of the model. It could be that the performance was not statistically different.

---

## Round 0.2 · Minor Revisions

A few minor revision are still requested.

Reviewer 1 ·

Basic reporting

The manuscript has seen improvements in language clarity, yet it still falls short in explaining "Alvin's methodology." The response provided by the authors indicates a misunderstanding of the concern, focusing only on the advantages of convolutional kernels over manual feature extraction. This response lacks specifics about what "Alvin's method" entails in the context of convolutional kernel design. The authors should clearly define the methodology, discuss its unique features, and provide relevant references. This will not only clarify its role in the model but also provide necessary context for readers unfamiliar with this approach.

Experimental design

The research question is relevant and fits well within the journal's scope. The authors have improved their explanation of the model's novelty, but there's a need to substantiate the model's lightweight nature. To do this, running the model on resource-limited machines for inference would be beneficial. Such tests would demonstrate its accessibility to users with limited computational resources, like small-scale farmers.

Validity of the findings

The comparison with baseline models shows impressive performance. However, the statistical significance of the 3-5% improvement remains unaddressed. The authors should explain why this improvement is significant, considering its relatively small margin. Is the improvement due to the innovative model design, or is it influenced by the dataset choice? Clarifying this will provide insights into the model's actual contribution to the field. Establishing the importance of this improvement will help in understanding its impact on practical applications, especially in agricultural settings.

Additional comments

The authors have made progress in addressing previous concerns, such as language clarity and data discrepancies. However, further work is needed to clarify specific methodologies and justify the significance of the findings. Providing detailed explanations of "Alvin's methodology" and addressing the statistical significance of performance improvements will enhance the paper's contribution. This clarity will ensure the research is both understandable and impactful, guiding future studies and practical applications in the field.

Reviewer 2 ·

Basic reporting

The authors have addressed my comments thoroughly, and I recommend the journal accept the paper.

Experimental design

The experimental design has improved based on the comments received.

Validity of the findings

The results have improved.

Additional comments

The paper is in good shape and recommended for publication.

---

## Round 0.3 · accepted · Accept

The paper can be accepted.

Reviewer 1 ·

Basic reporting

no comment

Experimental design

no comment

Validity of the findings

no comment

Additional comments

The author has successfully addressed my concerns.